# Brain Magnetic Resonance Imaging (MRI) as a Potential Biomarker for Parkinson’s Disease (PD)

**DOI:** 10.3390/brainsci7060068

**Published:** 2017-06-16

**Authors:** Paul Tuite

**Affiliations:** Neurology Department, University of Minnesota, MMC 295, 420 Delaware St SE, Minneapolis, MN 55455, USA; tuite002@umn.edu

**Keywords:** diffusion tensor imaging (DTI), fractional anisotropy (FA), functional MRI (fMRI), magnetic resonance imaging (MRI), magnetic resonance spectroscopy (MRS), neuromelanin, pharmacodynamic MRI (phMRI), resting state-fMRI (rs-fMRI), theranostics, voxel-based morphometry (VBM)

## Abstract

Magnetic resonance imaging (MRI) has the potential to serve as a biomarker for Parkinson’s disease (PD). However, the type or types of biomarker it could provide remain to be determined. At this time there is not sufficient sensitivity or specificity for MRI to serve as an early diagnostic biomarker, i.e., it is unproven in its ability to determine if a single individual is normal, has mild PD, or has some other forms of degenerative parkinsonism. However there is accumulating evidence that MRI may be useful in staging and monitoring disease progression (staging biomarker), and also possibly as a means to monitor pathophysiological aspects of disease and associated response to treatments, i.e., theranostic marker. As there are increasing numbers of manuscripts that are dedicated to diffusion- and neuromelanin-based imaging methods, this review will focus on these topics cursorily and will delve into pharmacodynamic imaging as a means to get at theranostic aspects of PD.

## 1. Background on Parkinson’s Disease (PD)

Parkinson’s disease (PD) is a neurodegenerative disorder which is associated with the accumulation of abnormally folded alpha-synuclein (a-syn) protein [1]. It has been hypothesized that PD progresses via prion-like spread of a-syn, which may follow highly connected fiber pathways that constitute a default mode network [1]. However, despite a-syn aggregation and development of Lewy pathology these changes are not always associated with neuronal loss, i.e., Lewy pathology and neuronal loss do not always overlap [2]. Thus, neuronal loss may be due to factors other than abnormal a-syn accumulation such as specific cellular susceptibilities [2,3]. Additionally, another crucial aspect is that neuronal dysfunction without cell loss may underlie many clinical features of PD. Thus, because neuronal loss, gliosis, Lewy pathology and neuronal dysfunction are some of the factors playing a role in the manifestations of PD, it is unlikely a single imaging method will diagnose, prognose, and stage PD. 

## 2. Imaging and PD

Presently there is no reliable positron emission tomography (PET) radioligand that is able to label intracellular a-syn in vivo in humans. Since there are many different forms of a-syn, e.g., monomers, dimers, oligomers, amyloid fibrils, etc., it would be important to determine what form of a-syn is labeled in order to understand how it relates to the pathogenesis of PD and to cell-specific susceptibility factors [2]. Meanwhile present imaging methods are unable to provide confirmation of subcellular pathogenic mechanisms of disease such as ER stress and oxidative stress. Nonetheless, non-invasive imaging methods may indirectly ascertain relevant aspects of pathogenesis by evaluating iron (which is a pro-oxidant leading to oxidative stress) and bioenergetic changes that relate to mitochondrial dysfunction [4,5]. Additionally, magnetic resonance imaging (MRI) can provide evidence of structural changes that occur as a result of the loss of dopamine neurons in the substantia nigra, alterations in myelinated fibers and gliotic changes, as well as the loss of non-dopaminergic neurons in other brain regions. Therefore, MRI methods that are sensitive to detect these tissue changes may prove useful as a biomarker. Additionally, improved resolution with enhanced signal-to-noise (SNR) can be achieved with a higher magnetic field strength, e.g., 7 Tesla (T) having enhanced SNR over 3T MRI imaging [6,7]. Because of its ready availability and non-invasive nature, many have utilized 3T structural MRI methods in their pursuit of developing a biomarker for PD; but, it is expected that with the increasing number of centers acquiring 7T platforms, there will be more data from studies with this higher field. 

To date, researchers have shown the discriminative powers of MRI in separating PD from controls by assessing myelin structure, free water, iron, fiber pathways, or an assortment of other aspects of diseased tissues that define PD [8,9,10]. With atrophy, there is an increase in perivascular space that may allow for a novel imaging approach [11]. Nonetheless, additional work is needed to show if structural MRI is able to diagnose PD in early stages, rather than when the disease is advanced and clinically definable from healthy controls. While this research has been primarily cross-sectional and requires a large number of subjects to show statistically significant differences, there has been some progress with the use of machine learning which may aid in the process to show cohort differences [12]. As well, longitudinal data has showed the feasibility of these methods to provide readout about “static” or slowly evolving changes that suggest the potential of MRI to monitor disease progression [5,13,14,15,16,17,18,19,20,21]. Another goal in developing an imaging biomarker would be to stage the disease in a variety of motor and non-motor domains, as well as to predict the future (prognosticate). To the former, longitudinal studies are beginning to better define the disease. 

Meanwhile, there has been some success in developing functional and neurochemical imaging approaches that may provide diagnostic abilities, as well as insight into evaluating treatment responses. This latter field constitutes pharmacological or pharmacodynamic imaging, e.g., pharmacological MRI or phMRI. Additional studies of these “dynamic” functional and “static” structural methods need to be done, as well as a determination of how they relate to one another. Specifically, some of these methods may be better in their ability to diagnose, stage, prognose, track, and evaluate responses to treatment, making it likely that a multi-modality imaging approach will be needed. This review will cover some insights into structural MRI biomarker research, but, will primarily focus on functional, particularly pharmacodynamic, MRI approaches. 

## 3. Structural Imaging

One way to consider structural imaging for PD is its ability to detect atrophy that occurs with the loss of neurons, axons, and large terminal arbors, along with gliosis, changes in myelinated fibers, iron deposition, and functional changes. When assessing for brain atrophy, many utilize T1-weighted segmented grey matter imaging data, which is analyzed using voxel-based morphometry (VBM) techniques. Atrophy ultimately occurs in PD; however, in previous studies of those with mild PD it has been controversial as to whether there is atrophy, no change or an increase in brain volume [22]. Detectable atrophy is more easily demonstrated in those with PD and mild cognitive impairment (MCI), or PD and dementia [22,23,24]. Specifically in these cohorts, researchers have shown atrophy in the basal ganglia, amygdala, hippocampus, parahippocampal gyrus, inferior/middle/frontal temporal gyrus, parietal and occipital lobes [22,23,24,25,26]. 

Meanwhile, others have employed a T1-weighted sequence that utilizes its sensitivity to neuromelanin to examine structures that have high neuromelanin content such as the substantia nigra and locus ceruleus [27]. Neuromelanin, which can bind iron, is altered in PD and thereby may provide a potential diagnostic test [28,29,30,31,32,33,34,35,36,37,38,39,40]. 

This review will also cover iron based imaging [41]. While iron deposition increases with aging, there are greater amounts of iron deposition in PD individuals, as compared to unaffected age-matched individuals. This has led to discussion as to whether iron is pathogenic or merely a marker of disease. Additionally, iron can be free or bound, and it may reside in neurons or glia. Increasing amounts of evidence support the role of iron-based imaging methods as providing a marker of neuronal loss. These findings may correlate with the clinical features of PD, but it remains to be determined if iron-based imaging methods are staging biomarkers [16,42,43]. Additionally, there is great interest in using iron-based imaging as a means to track experimental iron chelation treatment in people with PD, which is being administered in the hopes of altering the course of disease [44,45]. While researchers and patients await the results of a phase III clinical trial of the iron chelator deferiprone, iron-based imaging remains a solid but secondary structural imaging method. Other iron-based imaging methods, such as phase contrast data from susceptibility-weighted imaging (SWI), and quantitative susceptibility mapping (QSM) also demonstrate promise as staging biomarkers [46,47,48,49]. 

Another way to evaluate structure is to use diffusion tensor imaging (DTI). DTI is in a vast number of PD imaging manuscripts, including a 2008 study from our institution that showed structural changes in cognitively-normal PD subjects [50]. Recent reviews outline the presence of changes in early PD subjects, and the association between changes in fractional anisotropy (FA) and motor severity [19,51,52,53,54]. The review by Hall et al. delves into the search for DTI correlates to motor, cognitive, mood and other features such as hallucinations/psychosis, rapid eye movement sleep behavior disorder (RBD), and autonomic dysfunction in people with PD [54]. These authors highlighted the challenges whereby different DTI MRI methodologies are employed at different centers, as well as the need for a large cohort of mildly affected subjects to demonstrate structural changes in early PD. 

To answer this need, McGill University researchers used a standardized data set of 232 PD and 117 control subjects from the 3T MRI Parkinson’s Progression Markers Initiative (PPMI) to identify brain atrophy in early PD [55]. Using deformation-based morphometry (DBM) and tensor probabilistic independent component analysis (ICA), Zeighami et al. demonstrated atrophy affecting all components of the basal ganglia, the pedunculopontine nucleus, basal forebrain, an area containing the nucleus basalis of Meynert, the hypothalamus, amygdala, hippocampus, parahippocampal gyrus, and two thalamic regions in PD subjects [55]. As there were changes in numerous hub regions, these findings are in keeping with the hypothesis of PD spreading through connected structures. However, MRI is not able to specify the pathological substrate underlying imaging changes, i.e., is there neuronal loss and/or a-syn related Lewy pathology? The “atrophy” may actually be related to massive losses of arbors and axons rather than drop-out of cell bodies. MRI-pathological correlation studies in humans and animal models are needed to address this issue. As well, in the future concomitant a-syn PET imaging with MRI may help clarify the evolution of disease. Nonetheless, ongoing monitoring of these PPMI subjects with MRI will help determine the utility of imaging methods to provide a longitudinal marker of motor and non-motor features of disease. Meanwhile, research funding is aiding in the development of higher MRI field platforms with more sensitive MRI methods in order to reduce the needed number of subjects in studies, and to enhance our understanding of pathological substrates. 

Another imaging approach is magnetization transfer imaging (MTI) which focuses on exchange of magnetization between mobile and immobile restricted protons, and therefore, looks at magnetization between water and structures such as myelin or lipids in membranes [56]. MTI traditionally refers to magnetization transfer ratios (MTRs) rather than absolute quantitative measurements, the latter of which we and others have been developing [57]. MTI overall can provide information about neurodegeneration, as there are an assortment of pathological processes that may reduce the imaging signal [56].

## 4. Perfusion Imaging and Pharmacodynamic (Pharmacologic) Imaging

Phamacodynamic imaging can be used to evaluate drug effects, as well as appreciating clinical features of PD. Both MRI and PET imaging have implemented this approach in PD [5,58]. Traditionally ^18^F-fluorodopa (Fdopa) radiotracer administration with PET imaging has been employed to evaluate the integrity of the dopaminergic system, and when combined with raclopride (post-synaptic dopamine receptor radiotracer), it is possible to look at dopamine release in response to a variety of tasks [10]. Meanwhile, other PET methods use ^15^O-H_2_O to look at cerebral blood flow (CBF) or ^18^F-fluorodeoxyglucose (FDG) to assess brain metabolism. PET imaging has been used to evaluate PD subjects off and on symptomatic PD medications, and at rest or during a task, e.g., joy stick motor task [10]. Resting metabolic changes have been noted in PD using FDG PET methods whereby there is a pattern of reduction, i.e., PD-related pattern (PDRP) in the globus pallidus, putamen, thalamus, premotor and supplementary motor areas [59]. This may be due to functional changes, i.e., neuronal dysfunction rather than neuronal atrophy [59]. Another important FDG PET scan study showed changes in the primary visual cortex, which has relevance for metabolic MRI studies discussed below [60]. Additionally, researchers have demonstrated regional CBF (rCBF) changes in PD subjects in response to an oral levodopa challenge [61,62,63]. Building upon the understanding of dopamine as it relates to blood flow, researchers have used this paradigm to assess PD phenotypes. Specifically, researchers evaluated for functional differences in PD subjects who experience mood effects from orally administered levodopa, in comparison to a PD cohort of similar disease severity who did not have such mood alterations [64]. Using rCBF PET methods, they demonstrated changes in the medial frontal gyrus and posterior cingulate cortex (PCC), which presumably was related to abnormal dopaminergic modulation from caudate, anterior cingulate cortex or orbito-frontal cortex in those with PD and mood fluctuations. These findings may help in defining and managing clinical differences in patients. Another group demonstrated differences in connectivity between tremor-dominant from akinetic-rigid subtypes of PD using an on and off medication functional MRI (fMRI) tapping task [65]. While pharmacodynamic studies using oral levodopa can be done, there may be a delay in onset of action from this route of administration as well as variable bioavailability. A recent review demonstrating the use of intravenous (i.v.) levodopa in research suggests its viability as a safe, rapid acting, and potentially more reliable means to probe into pharmacodynamic effects using MRI or PET [66]. Thus, it is expected that pharmacodynamic studies using i.v. or inhaled levodopa (CVT-301) may help determine if this approach can be used as a diagnostic or staging biomarker. 

In regards to pharmacological MRI methods, there are two commonly employed options: blood oxygen level dependent (BOLD) signal changes and arterial spin labeling (ASL). 

## 5. BOLD

BOLD relies on the change in measurements of the amounts of oxyhemoglobin and deoxyhemoglobin in brain regions, which vary with neuronal activity [22]. BOLD changes can be ascertained when a subject is at rest, while performing a task, or to determine a therapeutic effect. Rodent and non-human primate PD models have been studied and have altered BOLD responses to medications, including the dopamine agonist apomorphine as compared to control animals [67,68,69,70,71,72,73]. Use of BOLD in people with PD is summarized in recent review articles [74,75], including work to evaluate treatment responses in people with PD [76,77,78,79,80,81,82]. A study by Mohl et al. 2017 used BOLD response to better understand the basal ganglia subcircuits in the postural instability/gait disorder- as opposed to tremor dominant-subtypes of PD [65]. 

Recently researchers have been focused on assessing for temporal correlations between BOLD from different brain regions, i.e., functional connectivity when subjects are at rest (resting state fMRI or rsfMRI) [74,75]. Using rsfMRI it is possible to evaluate PD and other synucleinopathies such as Lewy body dementia and idiopathic REM sleep disorder and to separate affected subjects from controls or other neurologic conditions [79,83,84,85,86,87,88,89,90,91,92,93,94,95,96,97,98,99,100]. Researchers have also focused on how functional connectivity changes in response to treatments, and how they relate to features of PD [86,87,101,102,103,104]. Typically, rsfMRI PD studies need to be done when subjects are off their usual antiparkinsonian medication in order to demonstrate differences between those with PD from healthy controls—and resumption of antiparkinsonian medications normalizes resting state abnormalities in PD [93]. Nonetheless, some researchers have found another functional imaging method, termed arterial spinal labeling (ASL), provides a more stable signal than BOLD [101].

## 6. ASL

ASL is an fMRI method that can provide insight into regional blood flow without the need for a nuclear radiotracer such as ^15^O-H_2_O, which is used with PET imaging [101,105]. Thus, ASL methods have been increasingly employed in evaluating psychoactive drugs such as ketamine, risperidone, lamotrigine, citalopram and amphetamine [101]. In PD, researchers have used perfusion-weighted dynamic susceptibility methods to measure rCBF changes with apomorphine administration in PD subjects [106]. There also have been an assortment of ASL studies in PD which have demonstrated blood flow changes such as a reduction in parieto-occipital and dorsolateral prefrontal perfusion [106,107,108,109,110,111]. One group showed correlations between regions where there were ASL perfusion changes with alterations in glucose metabolism using FDG PET on the same subjects; thereby validating ASL as a potential tool in PD research [111]. Using ASL investigators demonstrated a reduction in thalamic perfusion in PD subjects who were administered the symptomatic dopaminergic agent adenosine a2a antagonist syn115 (tozadenant) [112]. These ASL findings support the putative mechanism of action of this drug as well as assist in determining dosing or understanding its efficacy in patients [112]. Meanwhile, others have focused on ASL as a means to stage PD with a loss of a caudate perfusion laterality index in more advanced patients [113]. As part of the National Institute of Neurological Disorders and Stroke (NINDS) Morris K. Udall Parkinson’s Disease Research Centers of Excellence, one group has shown a correlation between rCBF and cortical thickness in a cognitively intact PD cohort providing a functional (ASL) and structural correlation of two different MRI methods that may be useful as surrogates for cognitive function [110]. The goal of understanding cognitive impairment with ASL methods was outlined in a prior study that showed, while there are typical cognitive differences between PD and Alzheimer’s disease, there are posterior ASL functional network similarities between these two conditions that support the idea of shared pathophysiological mechanisms [114]. In summary, given the seemingly greater reliability of ASL methods over BOLD, it is expected that there will be additional studies using these methods including pharmacodynamic studies in PD utilizing symptomatic treatments [101]. 

## 7. Neurochemical and Metabolic Imaging

Magnetic resonance spectroscopy (MRS) provides information about neurochemistry with (^1^H) proton and bioenergetics with phosphorus (^31^P) imaging. Our initial 4T ^1^H MRS study showed reliable neurochemical profiles in the substantia nigra (SN), but did not demonstrate the expected finding of a reduction in the antioxidant glutathione (GSH) in PD subjects as compared to controls [115]. The small size (2.2 milliliter, mL) of the SN region of interest (ROI) as well as the presence of iron and other factors, makes it difficult to know if there is selective nigral GSH reduction [115]. Our interest in GSH led to additional ^1^H-MRS work, where we utilized 8 mL occipital ROIs to measure GSH concentrations in two pharmacodynamic studies of the GSH precursor N-acetylcysteine (NAC): (1) a single (i.v.) dose study; and (2) a repeated oral dose trial [116,117]. The i.v. NAC study showed initial blood (200%–1000%) and then brain (~30%) increases in GSH concentrations [116]. Our 4 week daily oral NAC dose study in PD and control subjects showed blood antioxidant parameter increases (though 10-fold lower than the increases seen in the i.v. study), and trends, but not statistically significant, brain GSH increases [117]. The goal of the oral NAC study was to help determine dosing for a longer oral NAC clinical trial planned by colleagues at the University of California at San Francisco. However, our brain data did not assist us in creating useful dosing models. Finally, Mischley et al. has conducted an important study to evaluate the effect of intranasal administration of GSH on brain GSH levels as measured with ^1^H-MRS [118]. Their work shows promising findings, but is fraught with the challenges of placebo effects, and it remains unclear if the profound increases in brain GSH demonstrated are valid [118]. A second important finding we have discovered with ^1^H-MRS has been the ability to demonstrate changes in the inhibitory neurotransmitter gamma-aminobutyric acid (GABA) in the pons and striatum in PD, which probably relates to the manifestations (pathophysiology) of PD when subjects are off symptomatic medication [119]. It is expected that further ^1^H-MRS studies may explore the relationship of symptomatic PD medications on GABA, the relationship of GABA to disease symptomatology, and if GABA could be a staging biomarker, i.e., has a relationship to disease severity allowing for its use to track disease progression [119]. 

The second MRS method to be discussed is ^31^P-MRS, which provides bioenergetic information that could be used to monitor disease-modifying treatments. This review will outline the potential of ^31^P-MRS as it relates to an antioxidant clinical trial we are conducting. As it is well known that oxidative stress and mitochondrial impairments, with reductions in adenosine triphosphate (ATP), are thought to play a role in the pathogenesis of PD, many antioxidant treatments have been studied for their purported neuroprotective value for PD [120]. However, these trials have been unsuccessful. Thus, researchers have focused on creating models of disease that allow for greater insights into PD, as well as provide a platform to screen for newer treatments and evaluate their mechanism of action. Bandmann et al. has created genetic parkinsonian mutation fibroblast cell lines—one with a parkin and the other with a LRRK2 mutation—in order to screen for potential disease modifying therapies [121,122]. From these models, which have mitochondrial impairments and associated reduction in ATP, they found that the naturally occurring bile acid and Food and Drug Administration (F.D.A) approved medication ursodeoxycholic acid (UDCA, Actigal®) was able to rescue mitochondrial function and restored ATP levels to “normal” [121,122]. Given this promising data, as well as expertise at our institution in conducting a clinical trial of UDCA in a small cohort of patients with amyotrophic lateral sclerosis (ALS), we began to explore our ability to measure ATP with ^31^P-MRS and to look at the effects of UDCA on our measurements [123]. Meanwhile, others have conducted 3T ^31^P-MRS studies and have demonstrated a greater reduction in ATP and high energy phosphate in men than women with mild-moderately advanced PD, but did not find significant changes in a cohort of mildly affected PD subjects as compared to controls [124,125]. Our MRI group at the Center for Magnetic Resonance Research (CMRR) has demonstrated the viability of ultra-high field (7T) ^31^P-MRS methods to obtain reliable quantitative measurements of ATP, phosphocreatine, phosphoethanolamine, inorganic phosphate, glycerophosphocholine, and ratios of oxidized to reduced forms of nicotinamide adenine dinucleotide (NAD+ and NADH, respectively—the ratio NAD+/NADH representing the intracellular redox state) [5]. In our preliminary work using an 8 mL occipital lobe ROI, we showed lower cerebral ATP and NAD contents and altered NAD redox state, greater acidity, and reductions in precursors for phospholipid metabolism in PD subjects compared with controls [5,126]. Another important advancement that we have made is the development of an approach that can evaluate enzymatic activity of ATPase, which may be more dynamic and sensitive than absolute measurements of ATP itself [127]. We have now begun studies to evaluate two cortical ROIs simultaneously (frontal and occipital) to determine if both brain regions manifest ATP deficits, and if there are compensatory changes in ATPase. It is possible that these extra-nigral regions are feed-forward based with independent oxidative phosphorylation without regard to ATP levels [2]. These methods we have developed will be implemented in evaluating for pharmacodynamic effects of UDCA on bioenergetics in a small open-label high dose trial of PD subjects [121,123]. With this platform, it may be feasible to monitor bioenergetic impairments in PD in vivo and use these methods in staging and monitoring treatments. As there are several putative pathogenic mechanisms in PD and numerous different treatment strategies directed at different mechanisms, this approach may provide a downstream readout if such potential neuroprotective therapies are directly or indirectly impacting cellular energetics. Further technical advancements, such as other specialized head coils, would be required to obtain subcortical, e.g., striatal measurements. Given the small size of the SN, it seems ^31^P-MRS would face similar challenges that arose when we performed ^1^H-MRS imaging of this region, thus it is unlikely that ^31^P-MRS of the SN will be a feasible strategy for treatment monitoring.

## 8. Conclusions

In conclusion, diffusion and iron-based imaging may allow for staging PD; however, they remain to be validated as a diagnostic test. Pharmacodynamic/functional imaging through a variety of methods such as BOLD, ASL, ^31^P-MRS and ^1^H-MRS may provide insights into PD therapeutics and pathophysiology. Continued funding from federal and private sources are crucial for technological developments, as well as for evaluating MRI and PET methods, which will help determine their role as PD biomarkers.

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
