# Peer review of "Brain Magnetic Resonance Imaging (MRI) as a Potential Biomarker for Parkinson’s Disease (PD)"

_brainsci, 2017, doi:10.3390/brainsci7060068_

Round 1

Reviewer 1 Report

This is a review paper suggesting consideration of an MR imaging as a biomarker for Parkinson's disease. This review describes and updates current trends of imaging technology and imaging diagnosis methods for Parkinson's disease. It does not provide basic principles of the imaging technology, but it provides solid knowledge about multiple imaging methods and clinical application. Therefore, the reviewer suggests accepting this manuscript, since the focus of this manuscript is to introduce a recent advanced MRI technology as a non-invasive tool determining a biomarker for Parkinson’s disease rather than to deepen the knowledge about the principle of MR technology. Explanations and discussions in this manuscript will help our readers to learn a new technology and to broaden our knowledge about MRI application in a clinical environment.  

Author Response

Agree with comments

Reviewer 2 Report

This review is about the use of Magnetic Resonance Imaging to extract MRI-related biomarkers for the diagnosis of PD.

The review is well-written, the English of the manuscript does not require further revisions. The paper is structured in a clear way, and it is exhaustive, covering different imaging configurations.

However, as I am an expert of machine learning applied to medical imaging, I recommend the Editors to let the paper be reviewed also by a clinician, whose comments about the medical content would be more appropriate.

Comments

Lines 59-62: I would not say that machine learning is used to reduce the number of subjects required to show cohort differences. Does reference #12 show this? recently developed machine learning methods, such as Deep Learning methods, require (on the contrary) a huge number of sujects for training, in order to reduce overfitting problems. I would state that machine learning is used to perform automatic objective assessment, with some benefits including -among others- 1) the enhancement of classification/diagnosis performance, 2) the enhancement of sensitivity/specificity, 3) the reduction of the time required for the diagnostic process, and 4) the ability to deal with a huge amount of data, with respect to both the number of subjects and the number of features analyzed.

A review should only report results already obtained and published in scientific journals. In some points, the authors talk about their present work and they make inferences about possible results (e.g. lines 239-241).

The autors don't report any quantitative value about the analyzed studies. In particular, it would be interesting to see the accuracy (sensitivity/specificity) of different imaging configurations (thus, the accuracy of different biomarkers) in diagnosing PD.

The authors only make mention to the use of machine learning techniques for the extraction of MRI-related biomarkers. However, it would be very interesting to see if the use of these techniques could improve the performance of MRI in classifying PD (see, for example, the paper by my research group "Machine learning on brain MRI data for differential diagnosis of Parkinson's disease and Progressive Supranuclear Palsy"). A comparison between machine-learning methods and "classical" methods could be very useful.

Author Response

We appreciate the time by this reviewer to improve this manuscript and my responses are below:

I modified the wording for lines 59-62.

Discussion about unpublished data from lines 239-241 was removed.

Consideration for a statistically driven manuscript was made in the conception of this manuscript and since I am not a statistician and was unable to get our statistician to embark on this idea (I really tried) this approach was not used in this manuscript. I  completely agree that such a paper would be helpful in the literature but not as a review but an original manuscript that will require a fair amount of technical work in comparing imaging methods. Therefore I don't think I will be able to fairly address this comment without completely revising my work and finding someone to help on this initiative. I would be happy to work on such an effort in the future. 

I agree with the reviewer that a machine learning paper would also be something that should be for PD akin to the PSP manuscript but I don't think it is possible to do this easily at this time in the manuscripts maturation period. 
